# Antimicrobial Activity of Calcium Silicate-Based Dental Materials: A Literature Review

**DOI:** 10.3390/antibiotics10070865

**Published:** 2021-07-16

**Authors:** Ana Cristina Padilha Janini, Gabriela Fernanda Bombarda, Lauter Eston Pelepenko, Marina Angélica Marciano

**Affiliations:** Department of Restorative Dentistry, School of Dentistry of Piracicaba, State University of Campinas, Sao Paulo 13083-970, Brazil; anacristina_padilha@yahoo.com.br (A.C.P.J.); gabibombarda@yahoo.com.br (G.F.B.); lauterpelepenko@hotmail.com (L.E.P.)

**Keywords:** antimicrobial biofilm, bioactive materials, endodontics, root canal sealer

## Abstract

Endodontic biomaterials have significantly improved dental treatment techniques in several aspects now that they can be used for vital pulp treatments, as temporary intracanal medication, in definitive fillings, in apical surgeries, and for regenerative procedures. Calcium silicate-based cement is a class of dental material that is used in endodontics in direct contact with the dental structures, connective tissue, and bone. Because the material interacts with biological tissues and stimulates biomineralization processes, its properties are of major importance. The main challenge in endodontic treatments is the elimination of biofilms that are present in the root canal system anatomical complexities, as it remains even after chemical-mechanical preparation and disinfection procedures. Thus, an additional challenge for these biomaterials is to exert antimicrobial activity while maintaining their biological properties in parallel. This article reviews the literature for studies considering the antimicrobial properties of calcium silicate-based dental biomaterials used in endodontic practice. Considering the reviewed studies, it can be affirmed that the reduced antimicrobial effect exhibited by calcium silicate-based endodontic materials clearly emphasizes that all clinical procedures prior to their use must be carefully performed. Future studies for the evaluation of these materials, and especially newly proposed materials, under poly-microbial biofilms associated with endodontic diseases will be necessary.

## 1. Introduction

Endodontics in dentistry concerns the study of the morphology, physiology, and pathology of human dental pulp and apical tissues. This includes the normal biology, etiology of alterations, methods of diagnosis, preventive procedures, and clinical approaches for these treatments [1]. The growing demand for continuous improvements in the techniques and materials used in endodontics has been remarkable. In this sense, biomaterials have become a promising field of research and are now being developed to interact with complex biological systems and are mainly used in endodontic techniques involving dental perforation accidents, apexification treatments, and root canal filling after chemical-mechanical preparation [2]. The main reason for filling the root canal system is to seal the majority of its spaces, thus preventing the survival of microorganisms that interfere with promoting the forthcoming repair of the apical tissues [3], and eliminating any potential residual infection [4].

Among the endodontic materials indicated for root canal filling are tricalcium silicate-based materials as the main compound. The main advantages of these materials are related to their physicochemical and biological properties [5,6]. These materials have an alkaline pH after immersion, a high calcium ion release, and adequate flowability for endodontic use. Additionally, they can be considered as bioactive materials once they have a certain ability to induce the formation of hard tissue in both dental pulp tissue and bone; this encourages new treatment approaches for dentinal remineralization, vital pulp therapy, and bone regeneration [7] through the stimulation of cell proliferation and gene expression related to stem cell differentiation [8]. The potential antimicrobial properties of endodontic cements were previously attributed to their alkalinity and release of calcium ions [4].

Although considerable microbial reduction can be achieved after chemical-mechanical preparation, irrigation, and intracanal medication, the presence of bacteria in dentinal tubules and cementum after treatment still occurs, mainly due to the anatomical features of the root canal [9,10]. For this reason—especially when there is pulp necrosis and apical periodontitis—choosing a material with a certain level of antimicrobial activity can potentially help to reduce or prevent the growth of remaining microorganisms [11,12].

Primary infections in root canals contain microorganisms able to access and colonize the pulp tissue, impairing its function and leading to its necrosis [13]. Their microbial profile consists of several bacterial species that may lead to apical periodontitis once they reach the apical region. The most prevalent are *Fusobacterium, Porphyromonas*, *Prevotella*, *Parvimonas*, *Tannerella*, *Treponema*, *Dialister*, *Filifactor*, *Actinomyces*, *Olsenella*, and *Pseudoramibacter*. In addition, root canals with persistent/secondary infection are usually associated with post-treatment apical periodontitis, in which the first endodontic treatment has failed. The microbiota in these cases are composed of a group of species involving a predominance of facultative and Gram-positive anaerobic bacteria, such as *Streptococcus mutans*, *Streptococcus anginosus*, *Enteroccocus faecalis*, and *Staphylococcus aureus*. Therefore, the prevalence of biofilms is high, and clinically, one of their main characteristics is their greater resistance to antimicrobials [9,10,14,15,16,17].

Several in vitro studies have investigated the antimicrobial activity of endodontic materials through methods such as the agar diffusion test and the direct contact test [18,19,20,21]. Endodontic cements may have different inhibitory effects depending on their composition, as well as the evaluation method and selected test times. The direct contact test has been widely used to assess the antimicrobial effect of endodontic cements and root filling materials. The test is quantitative and is indicated for the analysis of insoluble materials and in standardized configurations [22].

This literature review aimed to investigate the published information regarding the antimicrobial activity of materials for root canal filling and reparative procedures with a calcium silicate-based composition used in endodontics.

## 2. Search Strategy

The literature search was performed on PubMed without language or year restrictions, according to the following search strategy:

(anti-infective agents OR antimicrobial AND biofilms AND bioactive materials OR calcium silicate-based dental materials OR biocompatible materials OR biomaterials AND endodontics AND root canal filling OR sealer OR repair material OR reparative endodontic materials OR hydraulic endodontic materials).

Duplicates were removed manually with help from a reference manager (Mendeley Desktop, software version 1.19.8). After the article screening, a manual search was conducted for the download of the complete texts. Other articles were then added by hand searching of grey literature (OpenGrey and Google Scholar). The details of the main outcomes classified by material type are shown in Table 1.

### 2.1. Inclusion Criteria

The PRISMA checklist was followed for this category of review; its workflow is shown in Figure 1. Studies of any design that analyzed the antimicrobial properties of hydraulic calcium silicate-based endodontic sealers and reparative materials, in vivo studies on both humans and animals, and in vitro studies conducted on any type of laboratory model were considered for inclusion in this review.

### 2.2. Exclusion Criteria

The exclusion criteria were as follows: articles in languages other than English, narrative reviews, experts’ opinions, and guideline reports. Studies were excluded if they evaluated the antimicrobial properties of other types of materials.

## 3. Root Canal Filling Materials (Sealers)

Antimicrobial activity is an expected feature of an endodontic sealer because these materials are used in contaminated clinical sites [34,35,36]. Gram-negative bacteria secrete virulence factors such as lipopolysaccharides, an endotoxin that potentially stimulates bone resorption by acting in the synthesis and release of cytokines, which in turn activate osteoclasts, thus being directly related to the occurrence of periapical lesions. Enterococcus faecalis is a facultative Gram-positive bacterium that has been associated with several oral diseases, including endodontic infections, apical periodontitis, peri-implantitis, and endodontic-periodontal diseases. For this reason, this microorganism is used in several in vitro studies to test the antimicrobial properties of dental materials and their effectiveness in endodontic procedures [16,37].

The antimicrobial properties inherent to endodontic cements, in some cases, are transient and rarely extend beyond 7 days, not being sufficient against persistent infections [28,38]. The use of a material with long-term antimicrobial capacity could be decisive for the success of an endodontic treatment, as it would help to reduce the residual microbial load and prevent the formation of new biofilms. Furthermore, the use of antimicrobial additives in cements can be advantageous with the addition of quaternary ammonium compounds such as benzalkonic chloride and cetylpyridinium chloride and may increase the antimicrobial effect of the materials [39,40].

The use of nanomaterials—due to their small particle size—also potentially offers a large surface area/mass ratio and an increase in chemical reactivity when compared to its correspondent original material [41]. The addition of nanoparticles to endodontic cements has increased the antimicrobial activity inside dentinal tubules and the consistency of the materials [42,43]. Chlorhexidine is an example of a substance used in its nanoparticulated form in endodontic cements, offering broad-spectrum antimicrobial action, which makes it effective against Gram-positive bacteria, Gram-negative bacteria, and fungi [29].

Endodontic sealers such as EndoSequence BC Sealer (Brasseler, Savannah, GA, USA, EUA), BioRoot RCS (Septodont, Saint-Maur-des-Fossés, France) and Endoseal MTA (Maruchi, Wonju-si, Korea) have been widely studied. These sealers contain an amount of oxide compounds known to potentially have antimicrobial activity, such as Al_2_O_3_, Fe_2_O_3_, MgO, Na_2_O, NiO, and SO_3_ [23]. Additionally, the potential antimicrobial activity of the endodontic sealers may be associated with their alkaline pH, release of Ca^2+^ ions, and formation of hydroxyapatite with the interaction of dentin, significantly reducing biofilm formation and viability [6,44]. These cements after hydration undergo a reaction forming calcium hydroxide [45], which is responsible for their biological properties [24,46].

Microbial recontamination in root canal fillings is directly related to the presence of voids between the filling material and the gutta-percha within the walls of root canals [47]. The amount of sealer between the root canal walls and the gutta-percha points is critical, as it should be as thin as possible, aiming to reduce the infiltration of microorganisms, especially after single-cone techniques associated with the use of endodontic sealers [48]. Especially in round-shaped root canals, this technique has shown favorable results in relation to filling capacity; however, in canals with a large buccolingual extension, the single-cone technique requires a greater amount of cement, potentially resulting in a greater presence of voids [23,49].

Another important aspect to be evaluated is the physicochemical properties of root canal sealers that can influence the bacterial contamination of root canals. A number of studies have shown that biomaterials that have high solubility require more time for their complete setting [27,50,51,52]. Thus, this high solubility can influence the quality of root canal treatment and increase microbial infiltration over time [33]. Calcium silicate-based endodontic sealers are currently commercially presented in ready-to-use or powder/liquid formulas. Pre-mixed cements depend on ambient humidity to initiate the setting reaction, while for powder/liquid cements, water is present in the formulation itself [53].

The most used antimicrobial test for endodontic cements is the direct contact test [18,22,31]. The use of the agar diffusion test for antimicrobial analysis of endodontic cements has been discouraged, as it reflects only the diffusion capacity of the tested material, and not its antimicrobial potential [34]. Therefore, it was replaced by the direct contact test, which gives more reliable results. However, direct contact tests do not take into account the presence of dentin and the potential effect as part of the complex nature of root canal anatomy or for biofilm formation [26]. However, recent modifications were introduced to assess the antimicrobial effect of materials under conditions that most resemble the clinical condition, those found in endodontic infections using viability staining and confocal laser scanning microscopy inside root canals [25,54].

Based on these results, it seems fair to affirm that preventing bacterial recontamination by sealing is an important feature that the sealer must provide. For this reason, the long-term dimensional stability must be considered when developing this type of materials, and its stability in clinical use should be evaluated in a variety of methodologies. The available sealer compositions per se do not possess a robust and significant antibacterial effect; however, the root canal chemical-mechanical preparation is undoubtedly still the most crucial step of endodontic therapy regarding the reduction of levels of bacteria and their sub-products.

## 4. Hydraulic Calcium Silicate-Based Reparative Materials

Reparative endodontic materials have been widely used since their development in the 1990s, with the first generation of mineral trioxide aggregate (MTA), mainly composed of calcium and silicate elements [55,56,57]. This patent described the origin of this material, which had a gray color, as being based on type I Portland cement partially replaced with bismuth oxide serving as a radiopacifying agent. After this patent, the first commercially available material emerged: ProRoot MTA (Dentsply, Tulsa, OK, USA). However, from a biological point of view, there were no studies at that time demonstrating the full potential that this material would present, which ended up being an inversion of the material development process, where the industry indicated the material emphasizing its sealing properties for clinical use.

Considering the aesthetic aspect of using this material, a white cement was proposed in a new patent on 25 July 2002 [58]. The reduction of the iron oxide concentration from the composition of ProRoot MTA—which resulted in a gray material—gave space to start the production of ProRoot MTA white; however, the radiopacifying agent based on bismuth oxide remained unchanged even in this new white composition [59,60]. A similar alteration was made with to Gray MTA Angelus (Angelus, Londrina, Brazil) which was renamed to white MTA Angelus, also reducing the concentration of iron oxide in its powder [61] but keeping the bismuth oxide as the radiopacifier agent. A second formula alteration around 2017 in MTA Angelus altered its radiopacifier from bismuth oxide to calcium tungstate.

However, later studies indicated that the interaction of bismuth oxide with the collagen present in dental structures, together with the irrigating solution used during endodontic treatment of root canal therapy, were the main reasons for tooth pigmentation [62,63,64]. These studies resulted in the replacement of bismuth oxide with other substances such as calcium tungstate, zirconium oxide, and tantalum oxide serving as alternative radiopacifiers in compositions such as Biodentine (Septodont, Saint-Maur-des-Fossés, France), EndoSequence BC RRM Putty (Brasseler, Savannah, GA, USA), MTA Repair HP (Angelus, Londrina, Brazil), and White-MTAFlow (Ultradent Products Inc., South Jordan, UT, USA) [32,65,66,67,68].

Currently, hydraulic calcium silicate-based endodontic materials have gained significant prominence due to their potential antimicrobial properties, alkaline pH, and bioactivity [4]. These materials have the ability to release calcium and hydroxyl ions in the surrounding tissue where they are applied, thus favoring the creation of a favorable environment for cell differentiation both in dentinal tissues and bone [36]. Currently, these materials are widely used in dental clinics—not only in endodontics—such as in the processes of pulp revascularization, repair of accidental or carious perforations, treatment of internal/external root resorption, pulp capping, and retro-filling in endodontic surgery [69,70,71,72,73,74].

The antimicrobial potential of reparative endodontic materials is directly related to their surface of contact, potential alkaline pH, and hydroxyl release [75], as these factors are directly responsible for damage to lipids, proteins, and DNA in the cell membranes of microorganisms [76]. Another antimicrobial mechanism of these materials is the presence of calcium in their composition, which reduces the presence of carbon dioxide in tissues, a molecule which is used by anaerobic bacteria, in addition to their alkaline pH, caused by the hydroxyl ions, which potentially also favors tissue repair [77].

The chemical compositions, as well as the crystalline phases, are of fundamental importance for the understanding of the physicochemical and antimicrobial properties of reparative endodontic biomaterials. In addition, the powder particle size before the hydration process varies widely depending on the materials, and the smaller the particle, the potentially easier it will be to mix and handle the material. The presence of particles with a diameter smaller than the dentinal tubules could potentially play an important role in the perforation sealing capacity, assuming that the smear layer and debris of the application site have been previously removed [35,78,79,80,81,82].

It is known that long-term antimicrobial challenges constantly occur after restorative procedures, and these clinical conditions may cause treatment failures [67]. An attempt to add an additional antimicrobial mechanism was the addition of a nano-hydroxyapatite capable of eliciting antibacterial activity on *Streptococcus* and *Enterococcus faecalis* [83]. The use of different species and methods of cultivation (aerobic and anaerobic conditions) when testing materials is crucial when prediction of its clinical behavior is intended.

Testing soluble materials—such as reparative endodontic materials—over agar plates seems inappropriate once solubility halos are observed, indicating possible misleading interpretations [34]. Previous studies reported similar observations when testing hydraulic endodontic materials in contact with agar [67,68,75], reporting a limited antimicrobial effect of the reparative endodontic tested materials (Figure 2—Adapted from cited reference [74]); other studies using other methods such as confocal microscopy in contact with dentin and a mature biofilm [84] also concluded that previous disinfection of the site to be treated with these materials is crucial and mandatory in order to expect a positive clinical outcome. Different storage methods for antimicrobial testing can also obtain varying results; in water, for example, ProRoot MTA showed higher antimicrobial activity than when aged in blood and exhibited significant antimicrobial activity reduction after 7 days [35].

Standardization of antimicrobial testing is crucial for the evolution of material research on different bacterial strains. A minimum of at least three specimens for each material/group should be tested, and three test replicates should be performed; additionally, they should be run by the same operator under the same laboratory conditions [85]. Other aspects such as the nature of the material, its chemical characterization, adequate sample size, the sample dimensions, and the sterilization method should also be broadly considered prior to the antimicrobial tests [30,86,87,88]. To date, no specific ISO standard is yet available for testing hydraulic endodontic calcium silicate-based reparative materials.

Regarding the results observed for the antimicrobial effect of reparative endodontic materials, it could be inferred that during the clinical use of these materials, the application sites must be thoroughly disinfected in advance, as the materials—similarly to the case observed for sealers—do not possess strong antimicrobial efficacy. Additionally, further studies with reproducible and standardized methods are necessary for further assumptions. Clinical long-term controlled studies considering both the success rates and analyzing the cause of failures regarding the use of these materials are utterly necessary to understand and improve endodontic materials.

The main limitation of this review is the fact that it did not provide details on the overall research strategy and several materials in both categories: sealers and reparative. However, a manual search of papers that discussed the standardization of these materials was included. Additionally, the lack of standardization (i.e., ISO) for calcium silicate-based materials is a crucial concern when developing, testing, and providing clearance for clinical use of these materials.

## 5. Conclusions

Long-term antimicrobial challenges can occur after endodontic and restorative procedures and can cause failures in dental treatment. The reduced antimicrobial effect exhibited by calcium silicate-based endodontic materials per se clearly emphasizes that all clinical procedures prior to their use must be carefully performed, aiming for exhaustive disinfection of the dental tissues. It cannot be expected that these materials will achieve bacterial reductions attributable to their properties (i.e., alkaline pH) once they are constantly challenged by infection and body fluid interactions that might cause failure of their antimicrobial or sealing properties in the long run. Therefore, it is necessary that future in vitro studies use greater methodological standardization for antimicrobial analysis of endodontic cements. Preferably, new studies are indicated to evaluate polymicrobial biofilms associated with endodontic diseases, as well as the addition of new compounds and formulations to optimize the antimicrobial effect of calcium silicate-based materials.

## Figures and Tables

**Figure 1 antibiotics-10-00865-f001:**
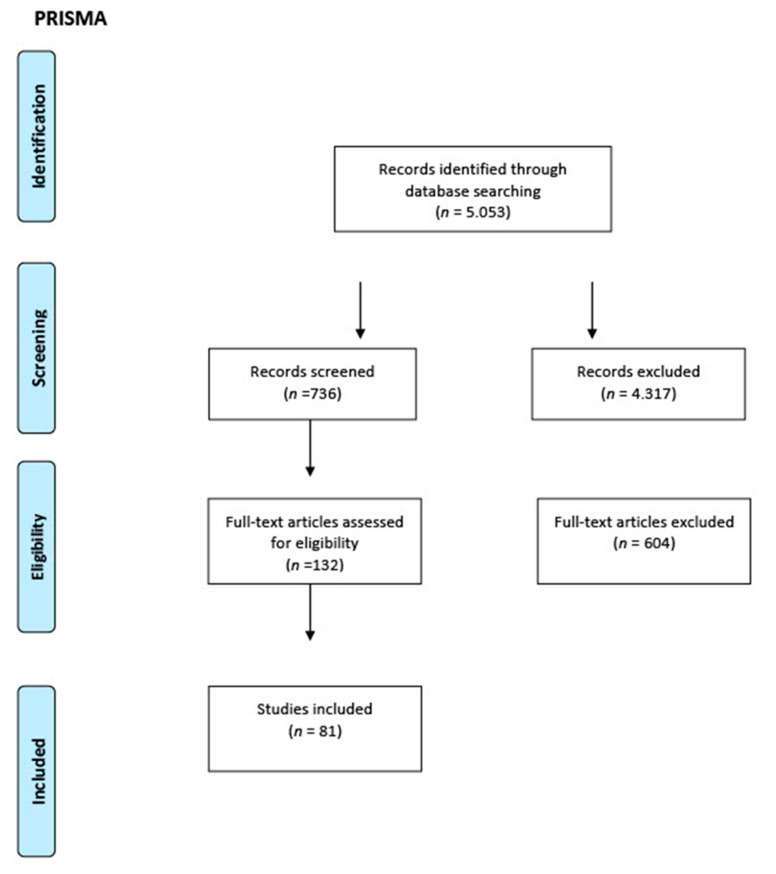
PRISMA workflow.

**Figure 2 antibiotics-10-00865-f002:**
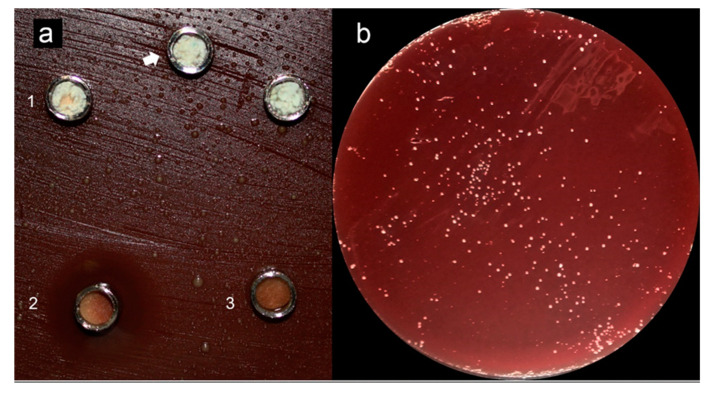
Adapted from [75]. (**a**) (1) Grey-MTAFlow cement without inhibition halos in BHI medium containing inoculated *E. faecalis*; (2) Chlorhexidine gel used as a control for antimicrobial activity presenting inhibition halo; (3) Additional metallic disc containing only silicon-based gel without inhibition halo as well; the arrow indicates the collection area for smear and viability tests. (**b**) Viability test for *E. faecalis* after 7 days showing viable bacteria adjacent to the disc. (**c**) The smear of *E. faecalis* after 7 days in contact with fresh Grey-MTAFlow cement. (**d**) The smear of P. gingivalis after 5 days in contact with fresh Grey-MTAFlow cement. (**e**) Representative SEM images of the surface of Grey-MTAFlow in contact with *E. faecalis* at 1000×, 2000×, and 5000× magnifications. (**f**) Representative SEM images of the surface of Grey-MTAFlow in contact with P. gingivalis at 1000×, 2500×, and 10,000× magnifications.

**Table 1 antibiotics-10-00865-t001:** Main outcomes reviewed, grouped in chronological order by material type.

Root Canal Filling Materials	
Author, Year of Publication	Main Outcomes
Zhang et al., 2009 [22]	Freshly mixed iRoot SP killed all bacteria within 2-min of contact, after 1 day of setting iRoot reduced the number of bacteria significantly during the first 2-min while all bacteria were killed within 20 min. IRoot had stable effectiveness for up to 3 days, but after 7 days it lost its efficacy.
Wang et al., 2014 [23]	All sealers killed more bacteria than the control group at all time periods. The antibacterial activity of Endosequence BC sealer increased over time. There was no difference between Endosequence BC sealer and AH Plus.
Candeiro et al., 2015 [6]	ADT: The inhibition zone of the AH Plus sealer was greater than in the EndoSequence BC sealer group. DCT: Endosequence BC sealer showed better effectiveness only after 24 h.
Arias-Moliz, Camilleri, 2016 [24]	ADT: MTA Fillapex revealed no antibacterial efficacy when exposed to water or PBS. In the EDTA group, MTA Fillapex showed the lowest antibacterial efficacy when compared with BioRoot RCS and AH Plus. CLSM: BioRoot RCS exhibited the greatest antimicrobial activity in all irrigation regimes followed by MTA Fillapex.
Alsubait et al., 2019 [25]	Antibacterial efficacy of AH Plus, Totalfill and BioRoot RCS was comparable after 1 day. Totalfill showed the highest number of dead bacteria after 7 days when compared to days 1 and 30. After 7 days, Totalfill killed significantly more bacteria than in the control group and BioRoot RCS. However, after 30 days of exposure, all sealers killed more bacteria than the control group, but BioRoot RCS killed a significantly higher percentage (61.75%) than Totalfill and AH Plus.
Bukhari and Karabucak, 2019 [26]	Endosequence BC Sealer was superior in killing *E. faecalis* compared with AH Plus at both time periods, 2 weeks and 24 h, with a statistically significant difference. There was no significant difference between 24 h and 2- weeks group within the Endosequence group.
Zordan- Bronzel et al. (2019) [27]	DCT: Totalfill reduced the number of *E. faecalis* significantly when compared with the control group. MDCT: Totalfill showed significantly higher effectiveness against *E. faecalis* when compared with AH Plus and the control group.
Barbosa et al. (2020) [28]	The results indicate that fresh Bio-C Sealer does not inhibit S. mutans growth, but exhibits antibacterial activity against *E. faecalis, S. aureus, P. aeruginosa* and *E. coli*.
Carvalho et al. (2021) [29]	The incorporation of chlorhexidine-hexametaphosphate nanoparticles can improve the antimicrobial performance of endodontic sealers.
**Hydraulic Calcium Silicate-Based Reparative Materials**	
Bhavana et al. (2015) [30]	All materials showed antimicrobial activity against the tested strains except for GIC on Candida. Largest inhibition zone was observed for Streptococcus group. Biodentine created larger inhibition zones than GICs, ProRoot MTA and GIC.
ElReash et al. (2019) [31]	Calcium silicate- based cements showed a potential antimicrobial activity mainly due to its high alkalinity (MTA HP and iRoot BP Plus). Antimicrobial effect of calcium silicate cements against strictly anaerobic bacterial species is still questionable.
Pelepenko et al. (2020) [32]	White-MTAFlow presents comparable antimicrobial properties to ProRoot MTA and Biodentine.
Queiroz et al. (2021) [33]	The pure tricalcium silicate associated with ZrO_2_, CaWO_4_ or Nb_2_O_5_ had appropriate physicochemical properties, antibacterial activity, cytocompatibility and induced mineralization in Saos-2, indicating their use as reparative materials.

## Data Availability

Not applicable.

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
