# Peer review of "Antimicrobial Activity of Calcium Silicate-Based Dental Materials: A Literature Review"

_antibiotics, 2021, doi:10.3390/antibiotics10070865_

Round 1
Reviewer 1 Report
The manuscript “Antimicrobial activity of calcium silicate-based dental materials: literature review” by Janini et al. covers recent literature on antimicrobial activity of calcium silicate-based dental biomaterials employed in Endodontics.
The review is well-organized, following an appropriate structure. The introduction is clear and concise. The literature search strategy (Section 2) was conveniently planned, allowing the selection of relevant articles on the desired topic. Remarkably, Table 1 is extremely useful to visualize the main outcomes of each reported article. Sections 3 and 4 are well-conceptualized and include a wide collection of studies. The conclusion section is a little bit short.
The development of novel endodontic biomaterials for efficient dental treatments is a hot topic over the last few years, so this review covering several aspects of calcium silicate-based dental biomaterials should be very exciting for the audience of Antibiotics. However, I have some specific suggestions that need to be addressed by the authors.
Comments to the Authors:
1) In the title, the authors should incorporate the article “a” before “literature”. Therefore, the title should be changed to “Antimicrobial activity of calcium silicate-based dental materials: a literature review”.
2) In the abstract, I recommend rewording “This article reviewed the literature…” to “This article reviews the literature…”. It seems more appropriate to use the present tense.
3) On page 6, “effective against Gram-positive bacteria, Gram-negative bacteria and fungi” should be reworded to “effective against gram-positive bacteria, gram-negative bacteria, and fungi”. In this case, “gram-positive” and “gram-negative” should be lowercase and hyphenated since they are used as a unit modifier. I also recommended to include the Oxford comma.
4) It would be valuable to incorporate some figures into the manuscript. These figures could highlight specific articles of particular interest or even summarize relevant aspects of the endodontic biomaterials.
5) The conclusion section is very brief. I recommend including some additional lines that help to better understand the content of the article. Additionally, it would be interesting to incorporate a short paragraph mentioning future perspectives on the field.
6) In its current state, the manuscript is easy to read. However, there are some small grammatical and syntactical errors. I suggest the authors reading the manuscript carefully and make corrections.
With all these minor aspects revised properly, I would recommend publication in Antibiotics.
Author Response
We kindly appreciate all the reviewer's comment about our manuscript. Following we response point-by-point all the suggestions:
1) In the title, the authors should incorporate the article “a” before “literature”. Therefore, the title should be changed to “Antimicrobial activity of calcium silicate-based dental materials: a literature review”.
The title was changed to:
Antimicrobial activity of calcium silicate-based dental materials: a literature review
2) In the abstract, I recommend rewording “This article reviewed the literature…” to “This article reviews the literature…”. It seems more appropriate to use the present tense.
The grammatical change was done and the phrase was rewritten to:
This article reviews the literature for studies considering the antimicrobial properties of calcium silicate-based dental biomaterials used in endodontic practice.
3) On page 6, “effective against Gram-positive bacteria, Gram-negative bacteria and fungi” should be reworded to “effective against gram-positive bacteria, gram-negative bacteria, and fungi”. In this case, “gram-positive” and “gram-negative” should be lowercase and hyphenated since they are used as a unit modifier. I also recommended to include the Oxford comma.
We apologise for the mistake. The sentence was rewritten according to the reviewer suggestion.
4) It would be valuable to incorporate some figures into the manuscript. These figures could highlight specific articles of particular interest or even summarize relevant aspects of the endodontic biomaterials.
We appreciate the comment and figure 2 was added with the proper referencing.
5) The conclusion section is very brief. I recommend including some additional lines that help to better understand the content of the article. Additionally, it would be interesting to incorporate a short paragraph mentioning future perspectives on the field.
We appreciate the comment and the ‘conclusion’ section was rewritten to:
- Conclusions
Long-term antimicrobial challenges can occur after endodontic and restorative procedures and can cause failures in dental treatment. The reduced antimicrobial effect exhibited by the calcium silicate-based endodontic materials per se clearly emphasize that all previous clinical procedures to their use must be carefully performed aiming for an exhaustive dis-infection of the dental tissues. It cannot be expected that these materials exert bacterial re-duction only attributed to their properties (i.e. alkaline pH), once these materials are constantly challenged by infection and body fluids interactions that might cause the material failure in the long run regarding either its antimicrobial or sealing properties. Therefore, it is necessary that future in vitro studies use greater methodological standardization for antimicrobial analysis of endodontic cements. Preferably, new studies are indicated to evaluate polymicrobial biofilms associated with endodontic diseases, as well as the addition of new compounds and formulations to optimize the antimicrobial effect of the calcium silicate-based materials.
6) In its current state, the manuscript is easy to read. However, there are some small grammatical and syntactical errors. I suggest the authors reading the manuscript carefully and make corrections.
We appreciate the compliments and apologise for the grammatical and syntactical errors. The manuscript was proof-read and the writing was double-checked entirely.
Reviewer 2 Report
This research is under the scope of this journal; the topic is relevant for readers, and this research deals with potentially significant knowledge to the field.
However, there are some concerns about the present manuscript:
Page 2: “Among the endodontic materials indicated for root canal filling are included the 40 tricalcium silicate-based materials as the main compound. The main advantages of these materials are related to its physicochemical and biological properties” Please consider references from 2021 (PMID: 33572611; PMID: 32651645)
Material and methods: Why PRISMA? Why not Systematic review?
There are many mistakes in the references section and in the text
The discussion is also misleading. What is the novelty of this paper???
Limitations?
Conclusions were not totally supported by the data showed.
Author Response
We kindly appreciate the positive comments regarding our manuscript. Following we response point-by-point all the suggestions:
Page 2: “Among the endodontic materials indicated for root canal filling are included the 40 tricalcium silicate-based materials as the main compound. The main advantages of these materials are related to its physicochemical and biological properties” Please consider references from 2021 (PMID: 33572611; PMID: 32651645)
The reference PMID: 33572611 was added to the citation. We appreciate the updating with the novel and relevant information of the suggested paper.
Material and methods: Why PRISMA? Why not Systematic review?
We appreciate the concern of the reviewer regarding the platform used. We shared the same concern when designing this review. PRISMA checklist is focused mainly in reporting systematic reviews and meta-analyses which surely was our first choice of review design; however, the heterogenicity of the studies and the lack of standardization – which is a crucial concern – for the reviewed topic resulted in several inaccurate search strategies which included some papers and excluded others given the advanced ‘all fields’ tool was used in PubMed. Additionally, manual search also included some important references of standardization of the material research which would not be included if we followed a strict search strategy for this topic. Therefore, we preferred to perform a ‘literature review’ addressed to emphasize both the limited antimicrobial effect of these materials and to suggest that a standardization was necessary for testing specifically calcium silicate-based materials.
There are many mistakes in the references section and in the text
The manuscript was proof-read and the writing was double-checked entirely, along with the references list.
The discussion is also misleading. What is the novelty of this paper???
Limitations?
We apologise for not including this crucial information. The last paragraph was included as follows:
The main limitation of this review is the fact that it lacked to provide details of the overall research strategy and several materials – in both categories: sealers and reparative – were not included at this moment. However, a manual search of papers that discussed the standardization of the materials were included. Additionally, the lack of a standardization (i.e. ISO) for calcium silicate-based materials is a crucial concern when developing, testing and providing clearance for clinical use of this materials.
Conclusions were not totally supported by the data showed.
We kindly observe that the conclusion was altered addressing additional points in this second version of the manuscript. Also, we observe that this topic was also a concern for the reviewer 1, and the rewritten topic is provided above.